# Identification of Prognostic Organic Cation and Anion Transporters in Different Cancer Entities by In Silico Analysis

**DOI:** 10.3390/ijms21124491

**Published:** 2020-06-24

**Authors:** Bayram Edemir

**Affiliations:** Department of Medicine, Hematology and Oncology, Martin Luther University Halle-Wittenberg, 06108 Halle (Saale), Germany; bayram.edemir@uk-halle.de; Tel.: +49-345-557-4890

**Keywords:** TCGA, human pathology atlas, gene ontology, organic cation transporter, organic anion transporter

## Abstract

The information derived from next generation sequencing technology allows the identification of deregulated genes, gene mutations, epigenetic modifications, and other genomic events that are associated with a given tumor entity. Its combination with clinical data allows the prediction of patients’ survival with a specific gene expression pattern. Organic anion transporters and organic cation transporters are important proteins that transport a variety of substances across membranes. They are also able to transport drugs that are used for the treatment of cancer and could be used to improve treatment. In this study, we have made use of publicly available data to analyze if the expression of organic anion transporters or organic cation transporters have a prognostic value for a given tumor entity. The expression of most organic cation transporters is prognostic favorable. Within the organic anion transporters, the ratio between favorable and unfavorable organic anion transporters is nearly equal for most tumor entities and only in liver cancer is the number of unfavorable genes two times higher compared to favorable genes. Within the favorable genes, *UNC13B*, and *SFXN2* cover nine cancer types and in the same way, *SLC2A1*, *PLS3*, *SLC16A1*, and *SLC16A*3 within the unfavorable set of genes and could serve as novel target structures.

## 1. Introduction

The organic cation and organic anion transporters belong to the superfamily of solute carrier (SLC) transporters and members are expressed in nearly all epithelia throughout the body [1]. The abbreviation of OAT, for organic anion transporter, is normally used for members of the SLC22 protein family. OAT1 for example is also known as SLC22A6 and OAT2 as SLC22A7. An overview about the nomenclature is given by Prof. Gerhard Burckhardt [2]. Members of the SLCO protein family are also called organic anion transporting polypeptides (OATP) [1]. In the same way, the abbreviation OCT, for organic cation transporter, is classically used for SLC22A1–SLC22A3 (OCT1–OCT3) [3]. Beside the SLC22A family, several other proteins are capable of the transport of organic anions and/or organic cations or are related with transport processes. Members of these protein families are expressed in nearly all epithelial cells. Physiologically, they are involved in the uptake and excretion of a broad range of substrates. For example, in liver and kidneys members of the organic anion transporter protein family are involved in the uptake of bile acid and the renal excretion of endogenous and xenobiotic compounds [2,4]. Further substrates include prostaglandins, steroid hormones, p-aminohippurate, monocarboxylates or acidic neurotransmitter metabolites, reviewed for example in [1,5,6].

Although involved in the transport of endogenous substrates, it has been shown for many members that they also transport xenobiotic-like drugs [6] and are thought to be involved in the intracellular accumulation of xenobiotic drugs [7]. Since they transport a wide range of substrates several studies have shown that members of the organic cation transporters and organic anion transporters are also capable to transport chemotherapeutics used for treatment of cancer, like platinum based chemotherapeutics, nucleoside analogs or kinase inhibitors [8]. Several studies focused on the identification of drugs that can be transported by a transport protein and its correlation to expression, treatment, and clinical outcome. It is important to know the expression pattern of members of the organic anion transporters and organic cation transporters in different tumor entities. The Cancer Genome Atlas (TCGA) provides next generation RNA-sequencing data for the most common tumor entities [9]. The data can be used to query the expression pattern of a gene of interest in different tumor samples. For many samples information also available includes clinical outcome, and the Human Pathology Atlas combined the gene expression level and generated a list of genes that are either favorable or unfavorable for clinical outcome of the patients [10]. So far, a systematic analysis of the prognostic value of organic anion transporters and organic cation transporters for the different tumor entities are missing. Here we made use of publicly available TCGA data to identify transporters that have either favorable or unfavorable prognostic value for the different tumor entities. We have also tried to identify common transporters that have a prognostic value in several tumor entities. The identified genes could serve as targets for the development of novel therapeutic drugs.

## 2. Results

The Human Pathology Atlas contains mRNA expression data from 17 different forms of human cancer. The expression data is derived from TCGA and correlation analyses based on mRNA expression levels in cancer tissue and the clinical outcome for patients have been performed to identify genes that are either favorable or unfavorable for overall survival of the patients. High expression of an unfavorable prognostic gene correlated with a poor patient survival outcome, and high expression of a favorable prognostic gene correlated with a longer patient survival. A prognostic gene for a given cancer was defined as a gene for which the expression level above or below the experimentally determined cutoff in an individual patient yields a significant (*p* < 0.001) difference in overall survival [10]. Appendix A shows the number of identified genes for different tumor entities derived from the Human Pathology Atlas.

The range in total number of prognostic genes goes from 57 (testis cancer) up to 5964 in renal cancer. Interestingly, the number of unfavorable genes in liver cancer is nearly ten times higher as the number of favorable genes. Unfortunately, the lists derived from the Human Pathology Atlas do not discriminate between the different cancer subtypes, for example in renal cancer, between clear cell renal carcinoma, papillary renal cell carcinoma, and chromophobe renal cell carcinoma. We have used the list to identify genes related with organic anion transport and organic cation transport that are prognostic for patients’ clinical outcome. To identify organic anion transport and organic cation transport related genes we used the Gene Ontology (GO) classification [11]. For the identification of organic cation transport, we used all genes that belong to the GO accession number GO:0015695 and for organic anion transport, all genes classified with the accession number GO:0015711. The list of genes were queried using the PANTHER classification system [12]. In total, 29 genes are classified as organic cation transport in the GO:0015695 and 453 as organic anion transport in the GO:0015711.

To get a more precise analysis we used the gene enrichment analysis to calculate if there is a positive (more than expected) or negative (less than expected) enrichment of a given GO for a given list of prognostic genes (Table 1) [12]. For the unfavorable list of genes, there is no significant enrichment of genes. However, there is a significant enrichment for favorable genes in the kidney (organic anion transport and organic cation transport), in lung and endometrial cancer (organic cation transport) and in liver cancer (organic anion transport).

The prognostic organic cation transport related genes for each tumor entity classified in the GO terms described above are shown in Table 2.

Interestingly, all the favorable prognostic organic cation transporters in renal cancer are favorable for overall survival for patients with clear cell renal carcinoma. *PDZK1*, *SLC22A2*, and *SLC44A4* have also a prognostic value for patients with renal papillary cell carcinoma. A similar pattern is also evident for the different lung and liver cancer subtypes. The organic cations only have a prognostic value for patients with lung adenocarcinoma but not for patients with lung squamous cell carcinoma. In the liver, the organic transporters only have a prognostic value for patients with liver hepatocellular carcinoma.

Since there are more genes classified as organic anion transporters, we used a column diagram to present the data. Appendix A shows the number of organic anion transporters that have a prognostic value for the different tumor entities. The list with gene names is provided in the Appendix A.

The number of genes more or less correlates with the total number of prognostic genes. In liver cancer the difference between favorable and unfavorable organic anion transporter organic anion transporter is smaller compared to total number.

Similar to the organic cation transporter, we have analyzed the prognostic value for the different renal, lung, and liver subtypes. Figure 2 shows the hazard ratio for organic anion transporters in the different renal cancer subtypes.

Out of the 96 favorable prognostic genes in renal cancer, 80 genes have a prognostic value for patients with clear cell renal carcinoma. Interestingly, *SLC4A2* have an unfavorable value for patients with clear cell renal carcinoma when analyzed separately. Out of the 96 genes, none has a prognostic value for patients with chromophobe renal carcinoma. In papillary renal cell carcinoma, 17 of the genes have a favorable and one gene an unfavorable prognostic value. The pattern for the unfavorable set of genes (69 in total) looks different. Only 25 have a prognostic value in clear cell renal carcinoma, while 7 of them have favorable prognostic value. In the chromophobe renal cancer cohort 10 genes and in the papillary renal cell carcinoma cohort 15 genes have a prognostic value.

A similar pattern is also evident for the liver and lung cancer subtypes. Most of the genes have prognostic value only for one cancer type (Figure 3).

From the 21 favorable genes, 17 have a prognostic value in the liver hepatocellular carcinoma cohort. Interestingly, the expression of *GOT2* has an unfavorable prognostic value in cholangiocarcinoma. Out of the 47 genes with an unfavorable prognostic value, 39 have a prognostic impact in the liver hepatocellular carcinoma (LIHC). None has a prognostic value in the cholangiocarcinoma cohort.

For lung cancer, 5 out of 7 genes have a favorable prognostic value in the lung adenocarcinoma cohort. None are prognostic for lung squamous cell carcinoma. Within the unfavorable set of genes, 6 out of 9 have a prognostic value in the LUAD cohort and none of them in the LUSC cohort (Figure 4).

In the same way as shown in Table 1, the lists of organic anion transporters were analyzed to identify a common set of genes separately for the unfavorable and favorable list of genes. Appendix A shows the number of genes that are common between the different tumor entities.

The number of prognostic genes is the highest in renal cancer (96 genes) and at least two (breast and lung cancer) and a maximum of ten (pancreatic cancer) genes are common in renal cancer. We have also identified genes that are common in more than two tumor entities. *UNC13B* is prognostic in five and *SFXN2* in four different tumor entities as shown in Table 3. The whole common gene list is provided as Appendix A.

While *UNC13B* and *SFXN2* are present in five and four tumor entities, respectively, only for renal cancer are both *UNC13B* and *SFXN2* favorable.

In the same way we have analyzed the lists of unfavorable organic anion transporters (Appendix A).

In renal cancer the number of prognostic genes (69) is the highest. Similar to organic cation transporters, the intersection between renal cancer and other tumor entities starts with 1 (breast) and goes up to 19 (liver) genes. One gene, *SLC2A1*, is present in five tumor entities (renal, urothelial, lung, liver and pancreatic cancer) and *PLS3*, *SLC16a1*, and *SLC16A3* are present in four tumor entities (Table 4).

With our approach we have identified organic anion transporters and organic cation transporters that are prognostic for a given tumor entity. We were also able to identify genes that are prognostic in several tumor entities. The highest numbers of genes were identified for the organic anion transporter. *UNC13B* is favorable in five and *SFXN2* in four tumor entities, implicating that these genes might have a general prognostic value. We have used the UCSC Xena platform to analyze if these genes have an prognostic value in the TCGA-PANCAN cohort with more than 12,800 samples derived from 17 different tumor entities [14]. We generated Kaplan–Meier plots and calculated overall survival probability for *SFXN2* and *UNC13B* (Figure 5).

The Kaplan–Meier analysis shows that high expression of *SFX2* and *UNC13B* are associated with a significantly longer overall survival probability. In the same way we have analyzed the unfavorable genes listed in Table 4 and calculated the overall survival probability in the TCGA-PANCAN cohort (Figure 6).

Similar to the favorable organic anion transporter, high expression of unfavorable organic anion transporter correlates with a poor overall probability in the TCGA PANCAN cohort. Unfortunately, the TCGA data does not provide information regarding the treatment strategy (medication used, etc.) for the individual patients. We have used the available information from the PANCAN cohort and analyzed if patients in the “treatment_outcome_first_course” indicated with “complete remission/response,” have a different expression of the above mentioned genes in comparison to patients qualified as “progressive disease” in the TCGA PANCAN cohort [15]. And indeed, the expression of *SFXN2* and *UNC13B* was significantly higher in patients with “complete remission/response” compared to patients with “progressive disease” (Appendix A). This was vice versa for *PLS3*, *SLC2A1* and *SLC16A1*, lower expression in patients with “complete remission/response” but higher in patients with “progressive disease”. There were no significant differences for *SLC16A3* (Appendix A). This data also shows that the expression level has influence on treatment outcome.

## 3. Discussion

Due to next generation sequencing (NGS) data from different tumor entities deposited in TCGA together with clinical data of patients, it is possible to identify genes that correlate with patients’ clinical outcome. The Human Pathology Atlas provides lists of genes where high expression is either favorable or unfavorable [10]. Highest number of prognostic genes is detected in renal cancer. One explanation could be that due to the different cell types involved in the physiological function of the kidney, a complex expression of genes is associated with each cell type [16]. Within this study we analyzed the prognostic value of organic anion and organic cation transport related genes. Regarding Gene Ontology classification, 29 genes are classified as organic cation transporters. While 7 of them have an unfavorable prognostic value, 17 have a favorable prognostic value, indicating that organic cation transporter gene expression is beneficial. Further support is given by the significant enrichment of organic cation transporter within the favorable set of genes in renal and lung cancer. No significant enrichment was observed within the unfavorable set of genes. For example, *OCT1* (*SLC22A2*) and *Mate-1* (*SLC47A1*) play important physiological roles in the proximal tubule of the kidneys [17]. Both genes are prognostic favorable in renal cancer and the high expression level might present less differentiation of the proximal tubule cells toward a tumor cell. This could be also an explanation for the prognostic value of *OCT1* (*SLC22A1*) expression in liver cancer. It is the main *OCT* in the liver and high expression could represent less differentiation toward a tumor cell.

The analysis of the different renal, liver, or lung cancer subtypes showed that the majority of the genes have a prognostic value only for one cancer subtype. In renal cancer most of the genes are prognostic in the clear cell renal carcinoma, in lung cancer in lung adenocarcinoma, and in liver cancer in liver hepatocellular carcinoma cohort.

In contrast to *OCT1* and *OCT2, OCT3* (*SLC22A3*) has a broad expression pattern [18] and is able to transport a variety of substances including drugs and chemotherapeutics and high expression has an unfavorable prognostic value in cervical and urothelial cancer. In contrast, for colon cancer patients’ receiving 5-fluorouracil, folinic acid, and oxaliplatin as therapy, the survival probability was higher when organic cation transporter-3 expression is high [19]. This was also evident for patients with head and neck tumors receiving cisplatin [20]. In both cases, high organic cation transporter-3 expression increased transport of the chemotherapeutic drugs. In pancreatic cancer, high *OCT3* protein expression correlated with better clinical outcome of the patients [21]. *SLC25A19* is the mitochondrial thiamine pyrophosphate transporter and mice deficient for this protein are embryonic lethal [22]. Studies showing an involvement in tumor diseases are missing.

In lung and renal cancer, we observe an enrichment of organic cation transporter in the favorable set of genes. For both cancer types, high expression of *MATE1* is favorable. *MATE1* is involved in the luminal excretion of organic compounds from cells that have been, for example, imported by basolateral expressed organic cation transporters. It has been shown that *MATE1* can transport a wide variety of endogenous and exogenous substrates (reviewed in [23]). *MATE1* is also able to transport cisplatin and to a higher content, oxaliplatin [24]. This would prevent accumulation of these drugs, for example in the cells of proximal tubulus, and thereby reduce the nephrotoxic effect of these compounds. On the other hand, reduced *MATE1* would have the opposite effect [23]. It has also been shown that *MATE1* expression is associated with better uptake of the tyrosine kinase inhibitor imatinib in chronic myeloid leukemia cells [25]. In this case *MATE1* has a beneficial role by providing sufficient uptake into the cells. There are studies missing that show a specific function of *MATE1* in treatment or progression of renal and lung cancer. We can only speculate if there is only a correlation of *MATE1* expression with patients’ survival or if there is a link with *MATE1* function.

Based on Gene Ontology classification more genes are classified as organic anion transporter related genes compared to organic cation transporter. But we have only an enrichment of organic anion transporter in the favorable set of genes in renal and liver cancer. Out of the classical *OATs* only *OAT2* (*SLC22A7*) has a favorable prognostic value in renal and liver cancer. These are also the main organs for *OAT2* expression [26] and similar to *OCT1* and *OCT2*, high expression of *OAT2* might represent less differentiation toward a cancer cell.

In contrast to organic cation transporter in renal cancer, some of the organic anion transport related genes are also prognostic for papillary and chromophobe renal cell cancer but the majority is prognostic for clear cell renal carcinoma. Interestingly, some of the unfavorable genes in renal cancer have a favorable prognostic value for patients with clear cell renal carcinoma. For example, high *SLC35A1* expression is favorable for clear cell renal carcinoma but unfavorable for papillary and chromophobe renal cell carcinoma. *SLC35A1* encodes for the CMP-Sia transporter and so far studies related to renal function are missing [27]. For the different lung and liver cancer subtypes, the majority of the genes are prognostic for one subtype, lung adenocarcinoma and liver hepatocellular carcinoma.

We have identified sets of genes that are prognostic in different cancer entities. For example, high expression of *UNC13B* is favorable in five and *SFXN2* in four different cancer types. Both genes together are cover nine different tumor entities. *UNC13B* also known as mammalian homologs of *Caenorhabditis elegans uncoordinated gene 13* (*MUNC13*) and it has been shown that it is involved in regulated exocytosis of vesicles [28]. *UNC13B* is not a classical organic anion transporter since it does not directly transport any substances. So far, no function has been described for *UNC13B* in cancer. *SFXN2*, or *sideroflexin 2*, is an evolutionary conserved protein that is expressed in outer mitochondrial membrane and is involved in iron homeostasis [29]. Similar to *UNC13B*, studies showing any functional correlation of *SFXN2* with cancer progression, developments, etc. are missing. Since the expression of both genes is associated with a significant better overall survival in the TCGA PANCAN cohort, further studies analyzing the cellular function of both proteins could be promising.

Within the unfavorable organic anion transporters, *SLC2A1* is present in five tumor entities. *SLC2A1,* also known as *GLUT1*, belongs to the members of glucose transporters which are involved in the transport of glucose across the cell membrane [30]. Since tumor cells have an increased metabolic rate, the expression of *SLC2A1* is often deregulated in different cancer types [31] and since they are involved in the support of tumor cells with energy, they present therapeutic targets [31].

PLS3, also known as T-plastin, is an actin binding protein and not a classical organic anion transporter [32]. High *PLS3* expression has been shown to have poor prognosis in pancreatic cancer, acute myeloid leukemia, gastric cancer, and colorectal cancer [33,34,35,36]. A direct function of PLS3 as transport protein is not known in the analyzed cancer entities.

SLC16A1, also known as MCT1 (monocarboxylate transporter 1) and SLC16A3, also known as MCT4, are involved in the proton coupled transport of lactate, pyruvate or ketone bodies and their predominant role is the transport of these substances in and out of the cell [37]. They can also act as drug transporters. MCT1 is able to transport valproic acid, nicotinic acid, nateglinide, and gamma-hydroxybutyrate across the plasma membranes [38]. They served also as targets for drugs. For example the inhibition of MCT1 activity in T cells inhibited effective immune response by reducing the rapid T cell division during activation process [39].

The unique pH characteristics in tumor cells provides development of treatments that target pH-related mechanisms to selectively kill cancer cells [40]. Cancer cells have a more alkaline pH compared to normal cells and this is even more evident in aggressive tumor cells [41,42]. This is mediated by transporters, like MCTs, that mediate proton transport out of the cells and might explain why high expression of *MCT1* and *MCT4* is associated with an unfavorable clinical outcome [42]. Targeting CD147, a MCT chaperone, by siRNA induced a decrease in *MCT1* and *MCT4* expression which was associated with reduced glycolysis, pH, and ATP production in melanoma [43]. MCT4 plays also an important role in cell migration [44] and targeting *MCT1* and *MCT4* expression has been shown to reduce the malignant potential of pancreatic cancer [45].

Similar to *UNC13B* and *SFXN2*, high expression of *SLC2A1*, *PLS3, SLC16A1*, and *SLC16A3* is associated with a significant reduced overall survival of the patients in the PANCAN cohort. While for *SLC2A1* this could be explained by the functional support of the tumor cells with glucose, there is no explanation if PLS3 is functionally associated with tumor progression.

Within this in silico analysis, we have tried to identify organic anion transporters and organic cation transporters that have a prognostic value for patients with different tumor types. This study however also has limitations. The Human Pathology Atlas provides data that show if the expression of a given gene is prognostic for a patient’s overall survival. The expression data is obtained at the time of diagnosis/biopsy/surgery and information about the individual treatment strategy is missing. It is important to analyze why, for example, *SLC22A3* expression is unfavorable. It has been shown that *SLC22A3* is involved in the uptake of cisplatin, and in head and neck or colorectal cancer upregulated *SLC22A3* expression improved cisplatin uptake and survival of the patients [20,46]. This one example shows that beside gene expression, also the treatment regime, including the used drugs, has to be included for a better understanding why a given organic anion transporter or organic cation transporter expression is prognostic and if there might be a functional interaction of the used drugs with the transporters.

It is also important to know if the observed expression of a gene derived from TCGA is derived from the tumor cell, the surrounding stroma cells or from infiltrating tumor cells. The cytotoxic T cells are the main tumor targeting cells and checkpoint inhibitors are widely used to keep activity of T cells at a high level [47]. The expression of *MCT1* is unfavorable in different tumor entities. Specific inhibitors could be used to down regulate the function of MCT1. As described above, MCT1 function is also important for the immune response of T cells. In this case, the functional inhibition of MCT1 could also counteract the T cell against tumor cell response. Therefore, it is important to know if a given deregulated expression pattern of a gene is mediated by the tumor cell itself or is derived from infiltrating immune cells.

## 4. Materials and Methods

For this study we have used free publicly available databases and online analysis tools. The lists of favorable and unfavorable genes for a given tumor entity were downloaded from the Human Pathology Atlas [10]. To identify organic anion transporters and organic cation transporters within the list of genes, the Gene Ontology classification was used [11]. For organic anion transporters, all genes classified in the GO:0015711 were used and for organic cation transporters, genes classified in the GO:0015695 were used. For classification, the list of genes were analyzed with the PANTHER classification system [12]. This helped to identify genes classified as organic anion transporters and organic cation transporters. In the next step, an enrichment analysis was performed to identify if organic anion transporters or organic cation transporters are either enriched or reduced in a given tumor entity [12]. The multiple list comparator was used to identify genes that are present in multiple tumor entities (http://www.molbiotools.com/listcompare.html). Survival heat maps and hazard ratio calculation were performed using the GEPIA2 web tool [13]. Single gene survival query using the TCGA PANCAN cohort and Kaplan–Meier analysis were performed with the Xena-Browser [14].

## Figures and Tables

**Figure 1 ijms-21-04491-f001:**
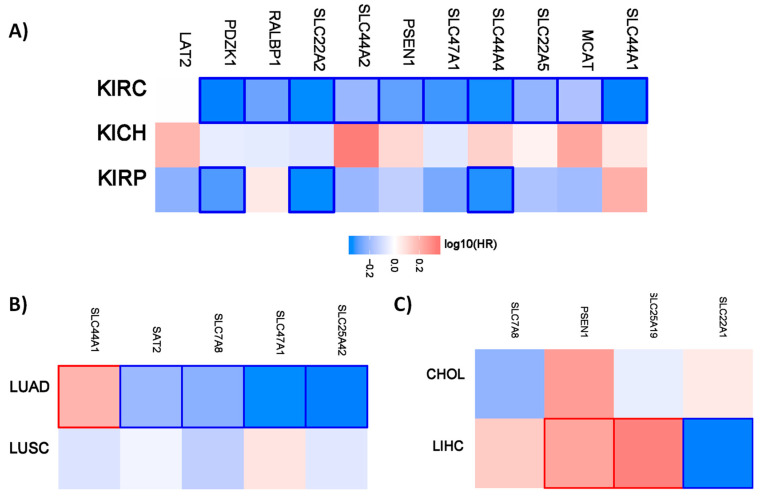
Hazard ratio of the organic cation transporter expression in different renal and lung cancer subtypes. The hazard ratio (HR) was calculated for (**A**) the different renal cancer (renal clear cell carcinoma (KIRC), renal papillary cell carcinoma (KIRP), and chromophobe renal cell carcinoma (KICH). (**B**) The hazard ratio for lung adenocarcinoma (LUAD) and lung squamous cell carcinoma (LUSC). (**C**) The hazard ratio for cholangiocarcinoma (CHOL) and liver hepatocellular carcinoma (LIHC). The red and blue color denote higher and lower risk, respectively. The rectangles with frames represent the statistically significant HR (*p* < 0.05).

**Figure 2 ijms-21-04491-f002:**
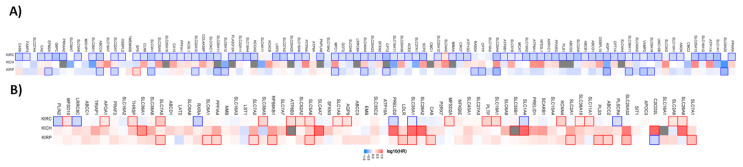
Hazard ratio of the organic anion transporter expression in different renal cancer subtypes. The hazard ratio (HR) was calculated for (**A**) the favorable and (**B**) unfavorable list of genes separately for renal clear cell carcinoma (KIRC), renal papillary cell carcinoma (KIRP), and chromophobe renal cell carcinoma (KICH). The red and blue colors denote higher and lower risk, respectively. The rectangle with frames represents the statistically significant HR (*p* < 0.05).

**Figure 3 ijms-21-04491-f003:**
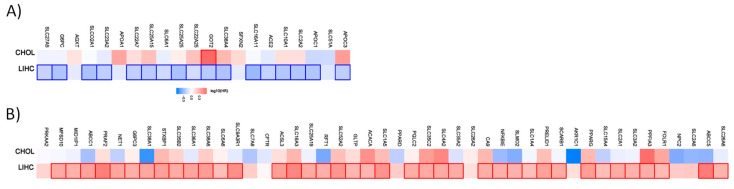
Hazard ratio of the organic anion transporter expression in different liver cancer subtypes. The hazard ratio (HR) was calculated for (**A**) the favorable and (**B**) unfavorable list of genes in liver cancer for cholangiocarcinoma (CHOL) and liver hepatocellular carcinoma (LIHC). The red and blue color denote higher and lower risk, respectively. The rectangles with frames represent the statistically significant HR (*p* < 0.05).

**Figure 4 ijms-21-04491-f004:**
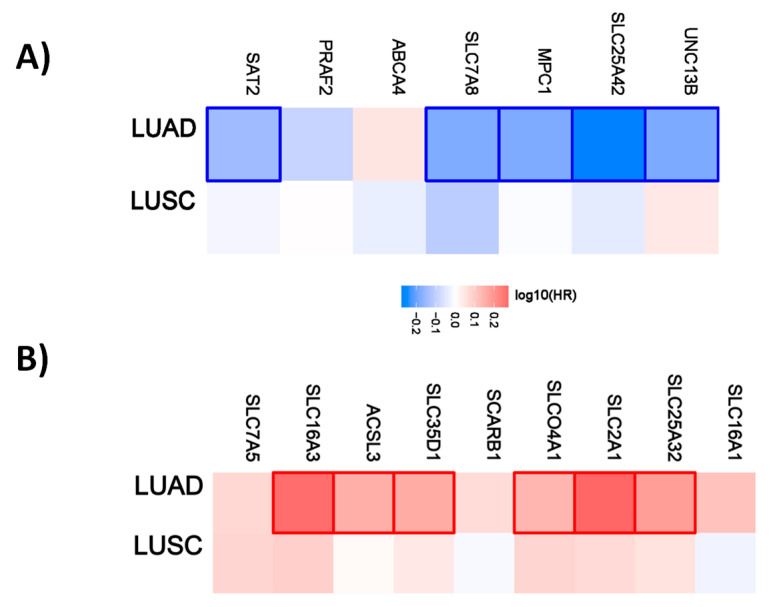
Hazard ratio of organic anion transporter expression in different lung cancer subtypes. The hazard ratio (HR) was calculated for (**A**) the favorable and (**B**) unfavorable list of genes in liver cancer for lung adenocarcinoma (LUAD) and lung squamous cell carcinoma (LUSC). The red and blue colors denote higher and lower risk, respectively. The rectangles with frames represent the statistically significant HR (*p* < 0.05).

**Figure 5 ijms-21-04491-f005:**
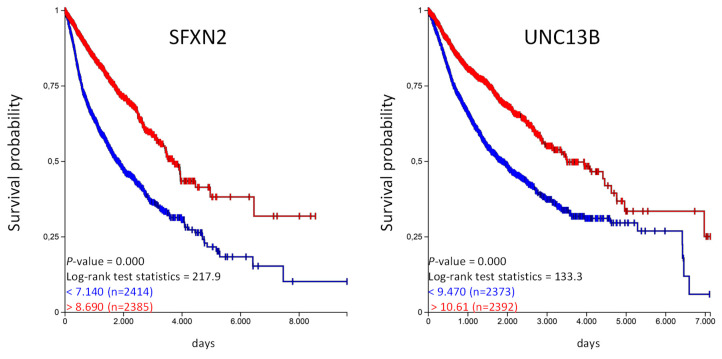
High *SFXN2* and *UNC13B* expression is associated with longer overall survival in the PANCAN cohort. The TCGA-PANCAN cohort was queried for *SFXN2* and *UNC13B* if they have an impact on survival probability. We filtered the data for primary tumor samples and generated Kaplan–Meier plots using the quartiles of gene expression level to separate high (red) and low (blue) expression level.

**Figure 6 ijms-21-04491-f006:**
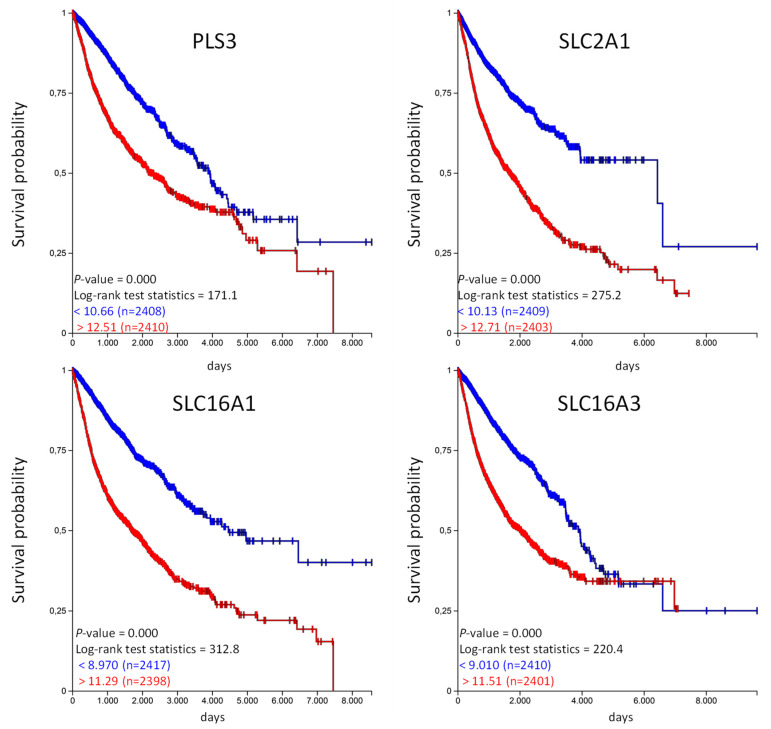
High *PLS3, SLC2A1*, *SLC16A1*, and *SLC16A3* expression is associated with shorter overall survival in the TCGA-PANCAN cohort. The TCGA-PANCAN cohort was queried for *PLS3*, *SLC2A1*, *SLC16A1*, and *SLC16A3* if they have an impact on survival probability. We filtered the data for primary tumor samples and calculated survival probability by Kaplan–Meier plots using the quartiles of gene expression level to separate high (red) and low (blue) expression.

**Table 1 ijms-21-04491-t001:** Enriched GO terms within the list of favorable genes. A significant raw *p* value (< 0.05) are highlighted in bold.

Tumor Entity	GO Biological Process	Fold Enrichment	Raw *p-*Value
Kidney	organic cation transport	2.57	0.02
	organic anion transport	1.58	0.00007
Lung	organic cation transport	9.43	0.001
	organic anion transport	1.06	0.8
Endometrial	organic cation transport	3.39	0.03
	organic anion transport	1.36	0.1
Liver	organic cation transport	2.65	0.3
	organic anion transport	3.57	0.0000009

**Table 2 ijms-21-04491-t002:** Prognostic organic cation transport related genes identified in different tumor entities. Genes present in two different tumor entities are highlighted in bold and genes present in three tumor entities are double underlined. Genes that are favorable for a given tumor entity but unfavorable for another tumor entity are underlined.

Tumor Entity	Unfavorable	Favorable
Breast Cancer	0	0
Cervical Cancer	***SLC22A3***	*SLC25A42*
Colorectal Cancer	0	0
Endometrial Cancer	***SLC25A19***	***SAT2*** *, **SLC47A1**, **SLC22A5**,* *MCAT*
Glioma	0	**0**
Head and Neck Cancer	0	***SLC44A4***
Liver Cancer	***SLC7A8*** *, PSEN1, **SLC25A19***	*SLC22A1*
Lung Cancer	*SLC44A1*	**SAT2,***SLC7A8,*SLC47A1, *SLC25A42*
Melanoma	0	**0**
Pancreatic Cancer	*SLC44A2*	** *SAT2, SLC22A5* ** *, SLC25A45, **SLC25A29***
Prostate cancer	0	0
Renal Cancer	*LAT2*	*PDZK1, **RALBP1**, SLC22A2, SLC44A2, PSEN1, * ** *SLC47A1, SLC44A4, SLC22A5,* ** ** *MCAT,* ** *SLC44A1*
Stomach cancer	**0**	** *MCAT* **
Testis cancer	***LAT2***	**0**
Thyroid cancer	**0**	**0**
Urothelial Cancer	***SLC7A8, SLC22A3***	** *SLC44A4* *, SLC25A29***
Ovarian Cancer	0	***RALBP1***

The number of organic cation transport related genes with a favorable prognostic value is higher compared to unfavorable group. The expression of *SAT2*, *SLC47A1*, *SLC22A5*, and *SLC44A4* are favorable in three different tumor entities. On the other hand, *PSEN1, SLC7A8*, *SLC44A1*, and *SLC44A2* are unfavorable in some tumor entities (liver, lung, or pancreatic cancer) and in others (renal and lung cancer) they are favorable. The expression of ten organic cation transporters is favorable in renal cancer and only the expression of *LAT2* is unfavorable. The Human Pathology Atlas does not discriminate between the different cancer subtypes. For example, renal cancer includes renal clear cell carcinoma (KIRC), renal papillary cell carcinoma (KIRP), and chromophobe renal cell carcinoma (KICH). We have used the GEPIA2 online tool to analyze the prognostic value of the organic cation transporters for the different tumor subtypes [13]. Figure 1 shows the results for the renal cancer and lung cancer subtypes.

**Table 3 ijms-21-04491-t003:** Top ten favorable organic anion transporters common between different tumor entities.

Gene ID	Present in
*UNC13B*	Renal, Lung, Head and Neck, Pancreatic, Colon
*SFXN2*	Renal, Urothelial, Cervical, Liver
*SIT1*	Head and Neck, Cervical, Endometrial, Melanoma
*MMAA*	Renal, Urothelial, Colon
*MPC1*	Renal, Lung, Cervical
*PITPNA*	Renal, Endometrial, Pancreatic
*PLA2G2D*	Cervical, Breast, Endometrial
*PRAF2*	Lung, Cervical, Pancreatic
*SAT2*	Lung, Endometrial, Pancreatic
*SLC16A11*	Renal, Liver, Pancreatic

**Table 4 ijms-21-04491-t004:** Top ten unfavorable organic anion transporters common between different tumor entities.

Gene ID	Present in
*SLC2A1*	Renal, Urothelial, Lung, Liver, Pancreatic
*PLS3*	Renal, Urothelial, Head and Neck, Pancreatic
*SLC16A1*	Renal, Lung, Endometrial, Pancreatic
*SLC16A3*	Renal, Lung, Cervical, Liver
*LDLR*	Kidney, Urothelial, Pancreatic
*SCARB1*	Kidney, Lung, Liver
*SLC15A4*	Kidney, Head and Neck, Liver
*SLC16A2*	Kidney, Urothelial, Breast
*SLC25A32*	Kidney, Lung, Endometrial
*SLC52A2*	Kidney, Cervical, Liver

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
