# Peer review of "Identification of Prognostic Organic Cation and Anion Transporters in Different Cancer Entities by In Silico Analysis"

_ijms, 2020, doi:10.3390/ijms21124491_

Round 1

Reviewer 1 Report

The article has been improved following the modifications that the author has made based on the recommendations of this referee. The confusing abbreviations OAT and OCT and the figures and tables with irrelevant data have been eliminated. The author has better explained the terms he used to refer to patient outcomes and the types and subtypes of cancer covered by each TCGA study.
The most confusing part that the author has not yet resolved in this version is the relationship between the expression of the carriers in the different types of cancer and the drug treatment that patients may receive in some cases. The author discusses the possible role that these transporters may have in relation to different anti-tumor drugs, but there are no results in the graphs to support this.
The author should specify well if the GCT data includes all types of patients, treated and untreated, and if he has included patients in the early stages of the tumor who do not normally receive chemotherapy. On some very relevant carriers, the author could investigate whether there is a relationship between patient OS as a function of carrier expression in treated and untreated patients.

Reviewer 2 Report

The author has improved his manuscript (ijms-789455) significantly. The introduction contains the essential background information, the figures present data in a clear format and main points are discussed in a well-structured order. The manuscript is now in a form to be accepted for publication.

Author Response

We are glad that we were able to answer all question.

Round 2

Reviewer 1 Report

The author has responded satisfactorily to the criticism of the previous version of the manuscript regarding the information available in the TCGA database on the different tumors and the drug treatment received by the patients.

This manuscript is a resubmission of an earlier submission. The following is a list of the peer review reports and author responses from that submission.

Round 1

Reviewer 1 Report

In this paper, the author conducts an in silico study looking for a relationship between the expression of a wide variety of carriers and the prognosis of many cancers in general. The study is confusing, with a poorly defined objective. It appears that most of the study is for preliminary data and ultimately focuses on some transporters and pancreatic cancer, without justification and with little depth.
Major concerns:
1) Using the abbreviations OAT and OCT for carriers of anions or organic cations, in general, is very confusing because these abbreviations are already commonly used to call different members of the SLC22 family. When reading the article, it seems that it is focused on these SLC22 transporters, but no, it is a group that includes all types of transporters.
2) The in silico study focuses on grouping genes according to their influence on the patient outcome as favorable or unfavorable. Although a reference is given, because it is an important parameter and is not clear to the reader, the criteria for defining favorable and unfavorable outcomes should be explained.
3) Many tumors that appear in the figures are classified as "cancer" in general. But most are carcinomas. The type of tumor is defined in each TCGA group and it is important to name it correctly. For some, such as liver cancer, hepatocarcinoma and cholangiocarcinoma appear separately in the TCGA. Have the authors gathered the information in the current study? The same is true for other cancers such as kidney, lung, or colon, where several groups appear on the TCGA.
4) The data represented in Figures 1, 2, 3, and 4, and Table 2 are not very relevant to the study. There is no justification for studying all types of cancer, and in the final and most interesting part, the authors focus on pancreatic carcinoma.
5) The objective of the study is not well defined. If these transporter groups are chosen it is to look for a relationship between the expression and prognosis of patients after receiving chemotherapy with drugs that are potential substrates of these carriers. However, no information is given in the study of the treatment received.

Minor changes
1) Line 36: it is said that transporters of anions or organic cations can transport drugs such as "platinum-based chemotherapeutics, nucleoside analogs or kinase inhibitors". To give these 3 types of drugs as examples is unnecessary because practically all the types of anti-tumor drugs used to meet this criterion.
2) In figures 5 and 6, time is represented on the abscissa axis. The units must be shown.

Reviewer 2 Report

The study by PD Dr Edemir (ijms-789455) identifies prognostic OCT and OAT candidates in various cancer types by using an computational approach on public data available in TCGA. The author finds some favourable but also unfavorabel candidate genes whichs expression is correlated with the outcome single or even a group of cancer types. The list contains well known candidats (SLC2A1 as positive control?) and also new ones. Although the central message of the manuscript is present, it leaves major points open (see comments). Therefore, the manuscript cannot be accepted for publication in the present form and needs major revisions.

Comments

The introduction should include a paragraph about the OCT and OAT transporters in more detail and examples of their substrates. Although the author cites corresponding review articles, it is not clear for the reader which role they play in cell physiology and how heterogenous this group of transporters is.

In the heading of Table 1 red marks are mentioned which are not visible in the current form of the table. The footnote of Table 1 should be included in the main text.

The message of the sentence in lines 36-38 is not clear, and the sentence should be revised.

Please exchange “hope” in line 50 by “postulate” or “assume”.

To support people with red-green-blindness, please exchange the red/green pattern in Fig. 1 and 2 by blue and yellow.

Why was the cut off for the cancer entities these 500 prognostic genes (line 64)? I would suggest to also include data from tissues with less candidate genes. Every single one counts.

Line 80: Please include “…we used a column diagram and not a table to present the data.”

Line 88: “The number of OCT and OAT related genes…”

Please describe the color code in Fig. 3 and 4.

What is the unit of the x-Axis in Fig. 5 and 6?

The discussion part is long, and it should be more structured to stress the main-message of the study. The sentences in lines 159/160 are not clear to me, because first the author writes about possible advantages of expression of unfavourable genes and the next sentence starts with “the disadvantage of the data…”. The text about MATE1 should be its own paragraph.

The text is full of typos and additional or missing words in numerous sentences. E.g. in the abstract in line 16 or in the Discussion in line 190 and 230. Please correct and revise precisely. The text contains 26 times “this”, please reduce.